# The Rise of Polymeric Microneedles: Recent Developments, Advances, Challenges, and Applications with Regard to Transdermal Drug Delivery

**DOI:** 10.3390/jfb13020081

**Published:** 2022-06-15

**Authors:** Aswani Kumar Gera, Rajesh Kumar Burra

**Affiliations:** Department of Electrical, Electronics & Communication Engineering, School of Technology, GITAM, Deemed to Be University, Visakhapatnam 530045, India; rburra@gitam.edu

**Keywords:** polymeric microneedle, polymers, skin, transdermal drug delivery, COVID-19

## Abstract

The current scenario of the quest for microneedles (MNs) with biodegradability and biocompatibility properties is a potential research area of interest. Microneedles are considered to be robust, can penetrate the skin’s deep-seated layers, and are easy to manufacture, and their applications from the clinical perspective are still ongoing with standard escalation. This review paper focuses on some of the pivotal variants of polymeric microneedles which are specifically dissolvable and swell-based MNs. It further explores the drug dissolution kinetics and insertion behavior mechanisms with an emphasis on the need for mathematical modeling of MNs. This review further evaluates the multifarious fabrication methods, with an update on the advances in the fabrication of polymeric MNs, the choice of materials used for the fabrication, the challenges in polymeric MN fabrication, and the prospects of polymeric MNs with applications pertinent to healthcare, by exclusively focusing on the procurable literature over the last decade.

## 1. Introduction

Continuous efforts and investigations are underway to develop a substitutional approach to the oral consumption of drugs and medications, for which the conventional method is hypodermic injections. The idea of the transportation of drugs through transdermal routes, wherein the pharmaceutics are transported to the skin’s innermost layers, was conceptualized and has garnered significant interest from the medical community [1]. One of the cardinal advantages is the mitigation of pain when compared to that of a traditional hypodermic injection. The other facet is the needle injuries that occur when the needle breaks after being pierced into the skin due to its diminished strength, and there is scarcely any chance of side effects; this differs from one patient to another [2]. Currently, in the commercial market there is an expansive range of gel-based solutions that can be applied to transdermal routes, including ointments and patch-based kits, which are available and are ready mixed to use on skin surfaces [3]. The skin’s structural area is primarily chosen depending upon the drug opted for and the rate at which the drug needs to flow and move into that particular area [4].

The transdermal means of transporting drugs through gels, ointments, and patches also have some hindrances in the delivery of their applications [3]. Explicitly, they have been found to be incompetent in delivering certain pharmaceutics through the deep-seated layers of the skin at a controlled rate due to the poor penetrability of the stratum corneum (SC) [5]. Therefore, multitudinous trials have been carried out to improve the penetrability of the drug using somatic methods and biochemical approaches, which have been consistently investigated from the previous decades. Chemical-based enhancers have shown good results in improvising the lipophilicity; however, the problems of skin irritability and low dosage exist [6]. There are other alternative methods, such as electrophoresis, iontophoresis [7], and the sonophoretic method [8], which disrupt the SC of the skin by suppressing the resistance capacity, preventing the penetrability of the drug. Nonetheless, the above methods are exorbitant and should meet the guideline frameworks and standards for an effective transdermal drug delivery.

In this regard, MNs are inexpensive, economical, and have a self-administrating type of TDD system for the delivery of various pharmaceutical drugs. They tend to permeate the stratum corneum (SC) to form transient microchannels that can dynamically diffuse micro-sized foreign molecules via the bloodstream [9]. MNs can also be adjusted or augmented to penetrate the skin layers without disturbing the nerves or hampering the blood vessels [10]. Metals, silicon, carbohydrates, and polymers amongst the other materials can also be deployed to produce MNs [11,12,13,14]. Microneedles that are produced from recyclable materials do not react with the medium and exhibit biocompatibility, such as polymers, which have numerous advantages due to their low cost, nontoxicity, and variety of physiochemical- and mechanical-related properties. Because MNs pass through biological barriers and come into contact with bodily fluids and tissues, the biodegradability and biocompatibility of polymer-based MNs (also known as polymeric microneedles) are discussed

Polymeric MNs’ capacity is expected to hike up as research on TDD in conjunction with polymeric MNs and their usage gains momentum. However, several challenges need to be addressed before these MNs can be used extensively in medical applications. Irritability of the skin, contamination on the micro-level and a lack of mechanical ruggedness in biomaterials, the volume of drug to be loaded on and in the MN, and the delivery of hydrophilic macromolecules are all challenges that need to be catered to [15]. Consequently, in the absence of accurate mathematical models of the drug release, determining the viability from the assortment of MNs available for targeting numerous diseases seems demanding and exacting. Scientific researchers have conducted mathematical modeling and simulation studies on polymeric MNs to add some value for optimization in the arena of drug delivery [16].

There is a significant need to cater to some of the primary and essential issues, such as the transformation from lab-based to user-based MNs, and their advancements towards product commercialization. This comprehensive-based review aims to discuss the latest techniques and developments related to the trials and challenges encountered by polymeric MNs in the areas of drug delivery routing mechanisms, polymer to polymer-based kinetics, type of materials used, and their variants with their evaluation techniques. Since the last decade there has been a sharp rise in the publications based on polymeric MNs, which implies the potential need for microneedles that are manufactured using polymers, which are graphically represented in Figure 1 below.

## 2. Polymeric Microneedles (PMNs)

Polymeric MNs have demonstrated their supremacy over silicone, metal, and other micro needles [17]. They aim to maintain both biodegradable and biocompatible properties and aspects such as cost-effectiveness and a wide variety of physicochemical and mechanical rigidity with fewer risk of material retention in the skin layers [18]. Polymeric MNs are further categorized as dissolving types of MNs and MNs that try to change its form (bulge) during the phase of metamorphosis. Regarding dissolving-type MNs, the drug is encapsulated on the tip of the microneedle. Dissolvable MNs absorb water dwelling in the skin and tend to completely soak into the skin, which results in the release of drugs to the inmost layers of the skin. At the base of swellable MNs, there exists a reservoir containing the drug in its lyophilized form. Swelling MNs absorb the available moisture in the skin, open the polymer-lattice-based matrix and then diffuse the drug via the attached reservoir, thereby making its way into the skin. Most swelling-based MNs are made of hydrogels, so the terms “swelling” and “hydrogel formation” are used interchangeably. As the needle array melts or softens, the disposal of medical waste is safe and free from the risk of puncture injury and contamination. Polymers exhibit all forms of bloating or swelling and degradability, and furthermore responsivity to stimuli of physical and biological elements. MNs which are derived from these polymers can control the physio-chemical and pharmacokinetic principles of drug-related molecules, as well as skin performance, in a variety of biomedical applications [19].

## 3. Polymeric Microneedles—Materials, Drug Transportation Kinetics

The choice of material for the MN manufacturing and properties of kinetics for the drug release will play a pivotal role in the transformation of MNs into commercialized applications for effective treatments for various ailments. The Georgia Institute of Technology, USA, and Alza Company, USA, have carried out preliminary pilot studies concerning the progress of MNs for the most effective medical treatments in the early period of the 1990s. Conventional metals or silicone were used in the early years as an opted-for material to manufacture of MNs [20]. As time passed by, the researchers started to use other types of materials that biologically gave them the option of viability. Encouragingly, they tended to be lighter and also showed promising signs of renewability [21]. Biodegradable polymer MNs were slowly developed in the early seasons of 2003 to check on the issue of biodegradability for the application of transdermal drug delivery [22]. Table 1 depicts the properties and fabrication techniques as seen below. 

### Polymer—Dissolution Kinetics

Kinetics is the study of the rate of change in levels of concentration. Dissolvability or dissolution means the course by which a substance alters its form from a solid state to an aqueous solution over time. Thus, dissolution kinetics is the study of the rate at which a substance’s concentration changes as it dissolves [38]. As a result, understanding dissolution kinetics is a critical component of modeling DMNs (dissolving microneedles). Several stimuli can dissolve the polymeric materials, which forms the dissolving MNs [39]. Here, we debate the core parameters that influence the performance of dissolving MNs, such as the temperature, UV, pH levels, and moisture responsivity of the polymers.

The term pH-responsive polymers refer to polymers that can allow a substance to be transferred into acidic or alkaline bases. Hollow-shaped MN arrays with microspheres encapsulating two pharmaceutical drugs along with sodium bicarbonate were produced by Ke et al. [40]. Protons could diffuse through the microsphere’s thin poly (L-lactic-co-glycolic acid) (PLGA). As a result, when the hollow MN array delivered the microspheres into the naturally acidic skin, the protons in the microspheres reacted with NaHCO_3_ (sodium bicarbonate), which is contained inside the microspheres. As an outcome, CO_2_ was formed, causing pressure to build up within the microsphere. The pressure was applied to the microsphere membrane until it was ruptured, allowing the encapsulated drugs to be released.

Moisture responsiveness is a term used to describe a material’s ability to absorb moisture. Polymers that are reacting for moisture responsivity dissolve due to the effect of hydrolysis. The moisture-responsive-based dissolving type of MNs can be designed either for an immediate or for a controlled release mannerism of a drug [41]. Koh et al. [42] provided the basic mRNA (messenger ribonucleic acid) using a polyvinylpyrrolidone-dissolving MN patch (RNA patch). The RNA patch preserved the physical and functional integrity of the condensed mRNA for a minimum period of two weeks. The kinetics of the RNA patch was almost the same as that of hypodermic injection when passed through the subcutaneous layers, leaves a fair good chance towards enhancing length of the MNs.

Enzyme responsivity of microneedles refers to the mode of sensitivity of the embedded drug in microneedles. Any variations or alterations in the biological signal are captured by these microneedles through application-oriented investigations and diagnostic purposes. The concerned pathologists and research community take advantage of the changes in the physiological signals by considering the parameters such as the overrated enzyme-based expressions, which ultimately trigger the release of the drug to the specific location.

In patients with diabetes, suffering from fluctuating blood glucose levels is faced and there is a dire need for them to keep themselves regularly monitored and cautioned about the baneful effects. In order to follow that, they must be administered with anti-diabetic medicines or anti-diabetic drugs through an injection. Until today, the most convenient and pain-free method of insulin administration is carried out with the use of glucose-responsive-based MNs. Glucose-sensing devices which follow the methods are discussed below [43,44,45].

Enzyme catalyzed reaction depends on the content of the glucose oxidase (GOx) as a primary catalytic enzyme primarily used for conversion of glucose to gluconic acid. The reaction will result in a substantial decrease in the potential of hydrogen(H_2_) and consumption of oxygen (O_2_) and thereby generate hydrogen peroxide H_2_O_2_ [45,46].Glucose(C_6_H_12_O_6_)-responsive-based devices are also prepared using other different kinds of materials known as lectins. The most common lectin which is widely in use is concanavalin A, which is a tetravalent-binding protein.Molecular-based recognition with the help of binding chemical moieties [44,45].

## 4. Polymeric MNs Design—Modeling and Optimization

MNs are miniaturized versions of the minimally invasive drug delivery type of devices, which necessitate an accurate mannerism and recurrent injection pricks on and into human skin layers [1]. Lutton et al. [47] suggested three primary necessities regarding normalized accepting criteria of MNs: they must (1) primarily permeate into the skin, (2) to permeate and to remain intact skin or even become dissolved and also intermingle with the therapeutic agent during drug delivery, and thereby (3) perform in the timeframe specified either to become dissolved in the bloodstream or detach from the skin [47]. Furthermore, to successfully insert MN devices of varying materials with their unique properties, with different geometries having varied array sizes, the effects of the design of the MN on the skin need to be evaluated [9].

Material choices are hypercritical, and they should be capable of controlling the manner of drug release dynamics and their stability during manufacturing for a safe and effective usage of MNs [34,48,49]. The materials (e.g., silicon, polymers, metals, carbohydrates) when arranged on top of the substrate (e.g., radial, triangular, square, or hexagonal), and geometries of the MNs (i.e., base diameter, tip diameter, base-to-tip ratio, and interspacing between centers) have an impact on the penetration depth of the MN [50,51]. To meet the FDA’s criteria for an MN system as a commercial medical product, it is critically important to control the length, sharpness, method of arrangement, and the puncturing rate or insertion rate of the MNs into the skin [52]. The ability of polymeric MNs to be inserted into the skin is affected by their shape, aspect ratio, and the tip radius of the MN [53], with an increase in the base width of the MN simultaneously with a compromise on the aspect ratio [54]. The failure force was observed to increase with narrowing of the tip and broadening of the base of the MN [55].

Chen et al. [56] revealed that pyramid-shaped MNs with the smallest aspect ratio stand out as those with the highest mechanical rigidity, and they can reach down to a good insertion depth. When observed, the mechanical strength of two MNs with the same aspect ratio with varying dimensions was found to be striking similar [56]. These observations of the researchers support the idea that their shape and aspect ratios are vital aspects when focusing on mechanical characteristics of MNs.

It is therefore understood that widening the MN bases, which simultaneously reduces the aspect ratio, would make skin insertion more challenging [54]. Sharpening the tip of an MN towards achieving for a good mechanical strength will result in a greater penetration depth into the skin [57,58]. MN insertion force decreases as insertion speed increases, whereas insertion force decreases as the MN tip center-to-center interspacing increases [59,60]. Polymeric MNs, having a pyramid shape with a lower aspect ratio and having sharp tips, achieve the best skin insertion rates [54]. Figure 2 below depicts the basic and essential parameters for design to evaluate the performance of polymeric MNs.

## 5. Microneedle Insertion—Behavior on Skin Layers

A wide range of factors influence the skin thickness, including the age, gender, ethnicity of the individual, body extent, and also hormonal status [61]. When MN patches permeate into the stratum corneum (SC), they surpass the layer which acts as a barrier, thereby delivering the full volume of the drug which is loaded in a quick and an easy manner [62]. The drug is then injected directly into the upper dermal layer, where it is dispensed for a systemic circulation, resulting in a pharmacological change every time the drug reaches the action site [63,64,65].

Because the thickness of the epidermis can reach up to 1500 m and MN lengths can reach up to a level of 1500 m, the drug is efficiently released into the epidermis. Most MNs range from 150 m to 1500 m long with a width from 50 to 250 m, having a thickness of the tip ranging from 1 m to 25 m [66]. MNs with a length of more than 1500 m can penetrate deep into the dermis, causing nerve pain and damage [66,67]. MNs only permeate the layer of the epidermis for the drug to be delivered, as shown in Figure 3 [60], whereas hypodermic needles can extend into the muscle. MNs with longer lengths can easily penetrate deeper layers of the skin, resulting in larger micro-sized holes. This allows more drugs to cross the stratum corneum (SC) to increase the permeability of the drug via the MN-created pathways [36,68,69]. Increased height of the MN can promote a vertical mannerism of drug diffusion while having a minor impact on the horizontal level of drug diffusion.

By adjusting the height of the MNs meticulously, the drug dosage can be precisely administered, which results in good drug distribution in the skin without any wastage [16,69]. MN patches with higher density can cause the drug to be diffused in a horizontal manner as the MNs create more micro-holes for a good dispersion via the micro-sized holes and penetrate into the layers of skin. As a result, the increased MN density is an operational method for increased drug penetration while considering large scales of MN skin pretreatment methods [70,71]. Figure 3 depicts the difference between a conventional needle and a microneedle.

## 6. Mathematical Modelling—Polymeric-Based Microneedles

The main goal of mathematical modeling is to achieve the optimal shape or desired design for microneedle optimization [72,73]. The development of mathematical models that simulate the physical mechanisms that occur during the process of transdermal drug delivery is important to the healthcare industry as it allows the quantification of critical factors which are found to be difficult to measure. When it comes to the skin, the stratum corneum (SC) is considered to be the chief barrier of volume transfer underneath the skin and percolating into its layers. When considering simulating the MN-oriented skin’s physiological system, an entire layer of epidermis needs to be considered [74]. Andrews et al. [75] clearly stated, with an evidential basis for the claim, that removal of SC significantly increases drug permeability. Additionally, if there is an exclusion of the entire epidermis, there is a good chance for increased drug permeability up to 1–2-fold. Basement membrane and tight junctions can resist mass transfer and should be considered in modeling [75]. Skin-associated parameters (e.g., porosity, the thickness of the skin, Young’s modulus, etc.) need to be considered in ineffective diffusion studies of MN-based therapies [76].

However, Sandrakov et al. [77] demonstrated conically shaped MNs are almost considered to be the most ideal, and it would be logical to begin modeling with a geometrical shape of a cone and making it the default. Generally, the MN performance is determined by prime factors such as the length of the MN, the tip of the MN, and its base diameter, interspacing between the two MNs, the total number of MNs when considering an array, and the distribution of MNs, be it (a square, in a diamond, or a triangle, a rectangle, or even an exceptional design), amongst other factors [16,78]. Another important aspect of modeling is the selection of physics, which allows the evaluation of drug release profiles. Fick’s law [8] is the governing equation for drug volume transportation. After deciding on the proper physics and defining all of the parameters, the model should be simulated using computational software with suitable computer hardware configurations. Commonly used computational software include the updated versions of MATLAB and also COMSOL Multiphysics [79,80].

Gomaa et al. [81] showed that the molecular weight affects the molecular diffusion rate of normal skin, especially when an MN pierces the skin. Zhang [82] developed a mathematical model for predicting the dynamic behavior of polymer electrolyte MN during drug delivery [82]. The mixing theory was used to classify the mathematical behavioral patterns of the skin (living tissue) as polyphasic liquid-saturated porous media, and conservative equations were used to clearly illustrate the response of the tissue through mathematical notation. The absorption of the drug through the capillaries and histiocytes was also mathematically modeled as a running medium that is found along the flow routes or paths [82]. Mathematical models have been used to solve the problem of drug absorption in capillaries and histiocytes. Zhang [82] worked towards innovation in drug delivery mechanisms. However, attention is shifting from fixed MNs to dissolved MNs.

Kim et al. [15] depicted their mathematical model to estimate the quantity of medication delivered through DMNs. In this model, an optimization algorithm technique was also used to work on the parameters that are exemplified by the data from the experimentation. According to Kim et al. [15], the algorithm which is used can be a tool for characterization of the drug-based delivery routines with dissolvable MNs. When focusing on the latest research on the advances in dissolvable MN technology, with the goal towards the healing effects of drug-related molecules, vivid thoughtfulness of the drug transportation mechanisms within the skin layers is highly recommended.

Ronnander et al. [83] established their mathematical model by contributing to the research on the evaluation of sumatriptan administration by using a restricted dataset. This mathematical model results in the simulation of the dissolvability of pyramid-shaped dissolvable MNs and the diffusion capacity, with the aid of governing equations. Chavoshi et al. [84] generated their mathematical-based model by replicating almost the same methods used by Ronnander et al. [76]. Chavoshi et al. [84], on the other hand, predicted drug release profiles using autocatalytic effects on polymeric dissolution. Chavoshi et al. [84] concluded that there are a lot of differences between the experimentation error values and the model-based error values.

## 7. Manufacturing of Polymeric Microneedles

Scientific and technological advancements have aided in the rapid production of versatile MN manufacturing techniques in recent years. The micro-molding method is considered to be an ideal fabrication method for manufacturing polymer-based MNs due to its high replicability, ability to easily scale up the production, and cost-effectiveness [85], with the advantages of processing at low temperatures, ease of the fabrication process, and above all, an almost negligible environmental impact. Moreover, the casting method is presently found to be the most popular method for the production of dissolvable polymeric-based MNs [86,87]. Droplet-born air blowing (DAB) is a futuristic kind of manufacturing method in which a droplet-sized polymer is formed in an MN by the process of blowing air [88].

Air, when being used directly for solidification of droplet-based polymers, helps in the development of the microneedle shape [89]. This droplet-born air technology is suitable for fabricating the microneedle under ideal settings, without the intervention of ultraviolet radiation or high temperatures [90]. A diversified range of pharmaceutical drugs are controlled using this technology by simply varying the amount of pressure and time through dispensers [91]. Figure 4 depicts the manufacturing process of dissolvable microneedles.

The procedure of drawing lithography involves creating microstructures that vary from a three-dimensional form to two-dimensional forms [92]. Above all, the conventional method of fabricating the MNs is possible only on a plain substrate surface, and it is a tedious task to fabricate on surfaces that are not in the plane on curvy surfaces [93]. In the last few years, three-dimensional 3D printing technology has evolved, and it is easing the process of manufacturing in an unprecedented manner. The core factor attributed to this technology is the tuning and flexible method, which facilitates the fabrication procedures.

This technology helps in achieving models according to the required geometrical shapes and sizes with the support of computer-aided designs [94,95,96]. This technology of prototyping has the basic fundamental technique of printing layer by layer, providing high accuracy with astounding resemblance to the model [97,98]. The current 3D printing techniques which are widely used by manufacturers and in the research community are the inkjet printers, the photo polymerization techniques, and the fused deposition molding for manufacturing polymeric MNs [99,100]. Figure 5 depicts the basic 3D printing technique process.

### 7.1. Emulsion and Bonding Fabrication Techniques

Emulsion techniques produce biodegradable microspheres, which can be easily be poured into a mold of MNs to form porous structures [46,101]. PLA has been commonly used to fabricate microstructures. The overall process of fabricated PLA as microstructures is shown through a pictorial representation in the figure below:

Step 1: The prepared PLA microspheres is poured into a female PDMS mold.

Step 2: It is pushed to the female-based cavities by the male structure of the PDMS.

Step 3: The pushing and casting process is repeated till the mold is filled with the microspheres.

Figure 6 depicts the inversion-based fabrication technique.

Step 4: Metal plate is commonly placed at the base of the mold and the top with a sheet of PDMS.

Step 5: The tip is pressed with the PDMS-based mold and ultrasonically welded. Heat is thereby generated when there is a friction with the microspheres, which ultimately yields a bond [46].

### 7.2. Micro-Molding-Based Fabrication Technique

Porous-based MNs with bio-ceramic materials are fabricated using these micro-molding fabrication techniques:

Step 1: Porous structures are prepared, and acidic-based solutions are blended thoroughly.

Step 2: The blended mixture is poured, then vacuumized, and finally cured in a female mold at a temperature of approximately 37 °C for two days.

Step 3: The MNs, which are interconnected pores, are produced by mixing water and acidic solutions, and they are cured at ideal conditions. The primary advantage of this fabrication technique is that the drug can be directly blended with the ceramic-paste-based materials and can be loaded onto the whole MN female mold [102]. Figure 7 depicts the fabrication process of porous-based polymeric microneedles.

## 8. Polymeric MNs—Challenges, Research Gaps, and Future Viewpoints

Even if the polymeric MNs are found to be biocompatible under the assumption of polymeric material accretion in the human or animal bodies when they are injected, they may cause impairment issues in the hepatic system accompanied by some detrimental immune-based reactions when these materials are inserted. To mitigate this, the insertion capabilities of the polymeric MNs need to be strengthened to make sure that they will not suffer due to breakage and bending issues after being inserted into the skin tissues [103].

To tackle the challenge, an amalgamation of a minimum of two or more polymeric materials is to be prepared to alleviate the structural dynamics and mechanical robustness of the polymeric MN [52] simultaneously to see that there will be a good balance of flexural strength when working with soft-based tissues, which fail to withstand high pressures and high rigidity during the insertion process of the polymeric-based MN. We would like to showcase and project what the future holds in the area of polymeric microneedle technology, emphasizing the wide scope for next-generation methods of processing and manufacturing these microneedles according to the requirements of additive-based manufacturing techniques and the usability of COVID-19 testing strategies and safe means of inoculation procedures. There are a good number of transdermal products which are facilitated by MNs; however, there is a wide scope for development and commercialization. However, certain hurdles need to be catered to in the areas of manufacturing of these microneedles and their cost-effectiveness. In an era of smart, mini-, and micro-based products, rapid development of polymeric microneedle devices is emerging for the betterment of living standards, as these devices are truly considered to be game changers.

### 8.1. Next Generation of Microneedle Technology

Several studies have been conducted through in vivo means of fabrication of MNs, which is used to transport pharmaceutical drugs and vaccines. The daunting task is to fabricate the MN, which can efficiently transport the macro-sized molecules with good molecular loads and a high amount of hydrophilicity [52,103,104,105]. There are varieties of derma rollers available in the commercial markets. However, to date there is no record of biodegradable polymeric-based microneedles on the global markets, and also no record of polymeric MNs, including protein-based products, on the market.

### 8.2. Microneedle- and Additive-Based Manufacturing

As discussed previously, the concept of 3D printing is being exercised vigorously by the research community and pharmaceutical and biotechnology-based industries. The usage of this technology stands out as promising when compared to that of the conventional methods of manufacturing, in terms of the amount of time consumed for manufacturing and cost-effectiveness [106,107,108].

### 8.3. Microneedle Impact on COVID-19

The microneedle found its way to tackle the global pandemic issue of COVID-19. The research team—Chen et al.—worked out and presented MN-based oropharyngeal swabs which substantially reduced the false negatives in testing procedures [109]. This concept greatly helped the doctors and testing personnel to identify the difference between a positive sample and a negative sample. If the vaccines are incorporated into these microneedles, people can easily administer the vaccine themselves, lessening the risk of long periods of expose during inoculations at vaccination centers and making their lives easier.

## 9. Applications

In the last decade, MNs have garnered keen interest from the research and industrial expert communities for their advantages in the area of transdermal drug delivery-based applications, which have been diversified and to add to its feathers more recently the inoculation delivery methods, diagnosis of diseases, and cosmetics in the commercial industry are gaining more application based interests.

### 9.1. Drug Delivery Applications

A very primitive application of these MNs was employed in drug transportation with the use of solid-based silicon MN [110]. A dissolvable MN patch that was diffused in caffeine is used for treatment in the reduction in obesity in mice that are overweight [111]. MNs are also used for the transportation of pharmaceutical drugs which are used as common pain killers such as ibuprofen and paracetamol [112]. In many studies and investigations, a microneedle-based array has been employed for transportation of drugs into humans and animals through the skin layers of human, mice, and porcine, and demonstrations were also performed on chicken breast and thigh [113] areas and even on tissues linked to the brain [114].

### 9.2. Inoculation Process—Delivery of Vaccine

In most cases, a dissolvable type of MN is considered as the best viable option for delivering the vaccine. In the past few years, the conventional hypodermic needles were commonly used for administering the vaccine, which is getting replaced with the dissolving type of MNs, which exhibit decent biocompatibility, appear to be robust with scalable properties, and are eco-friendly since they tend to cause the least amount of bio hazardous wastage [115]. To their credit, these dissolvable MNs have successfully delivered the vaccines for various diseases, such as malaria, diphtheria [116] influenza [117], hepatitis B [118], human immune papilla virus (HIV) [119], and also polio [120]. In addition to the dissolvable MNs, coated kinds of MNs are added to serve the purpose of inoculation [120,121]. In another study, hepatitis C virus (HCV) protein was successfully encoded in a deoxyribose nucleic acid (DNA) vaccine, which was coated on the top of the microneedle [120]. Figure 8 depicts the various microneedles devices.

### 9.3. Diagnosing and Therapy—Diseases

Diagnosing diseases and determining the effectiveness of therapies can also be achieved by looking into the biomarkers which are present in the body for evaluation of health. The existing procedures tend to be painful for the patient undergoing the test and require some personalized equipment [122]. Nevertheless, with the advent of microneedle technology, which deals with bioassays, the procedure has become simple with less intervention of equipment [122] It also helped to diagnose several other diseases, such as diabetes [37] and Alzheimer’s [120] disease, with much ease. In monitoring the health of a patient, a glass-based hollow MN was used to probe an amount of glucose into the body [44]. Nanoparticles also play a pivotal role when they are incorporated into microneedles, helping in the identification of biomarkers at a preliminary stage of osteoarthritic issues [123]. Figure 9 depicts the ocular drug delivery process.

### 9.4. Cosmetic Industry

Microneedles are thought to be an effective cosmetic treatment for aging, skin lesions, vulgaris, and wrinkles [124]. Kim et al. created a dissolvable MN patch based on hyaluronic acid for intradermal delivery of ascorbic acid and retinyl retinoate [120]. Kumar et al. demonstrated in vitro and in vivo enhancement of local delivery of eflornithine (used to reduce facial hirsutism) using a solid MN [124]. Furthermore, the technology was used to treat a couple of persons who suffered from alopecia areata (AA), which is an autoimmunity-related disease, and they expressed that they had significant growth of hair [125].

## 10. Conclusions

This paper depicts polymeric microneedles as a cutting-edge technology in the biomedical and pharmaceutical fields and presents a diversified scope for the future science and research areas. There has been remarkable and progressive research in transdermal routes of drug delivery through polymeric microneedles in the previous decade. This comprehensive review sums up the MNs technologies in the transdermal drug delivery era with regard to polymers and their types of materials, drug transportation kinetics, insertion mechanisms with mathematical modeling of polymeric MNs, and advances in fabrication methods, which are explored vividly. In the last sections, this paper even depicts the underlying gaps and the existing challenges which need to be met in MN research to realize an efficient means of low-cost fabrication. Finally, the paper wraps up with polymeric MN applications in diversified areas.

## Figures and Tables

**Figure 1 jfb-13-00081-f001:**
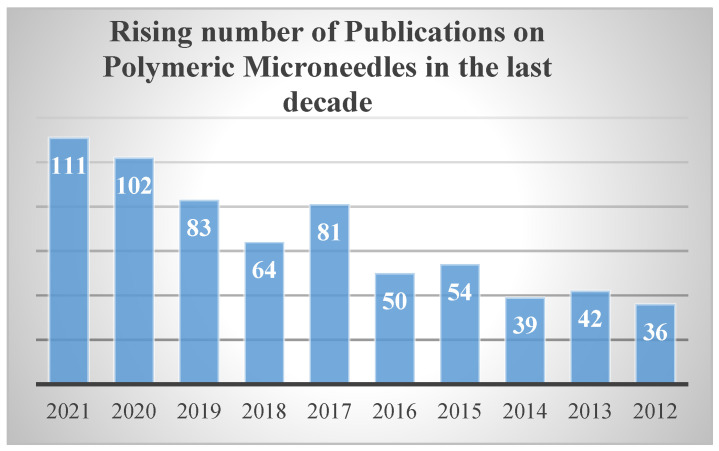
Publications on polymeric micro needles in the last decade. The above bar chart displays the incremental rise in published work on polymeric MNs over the last 10 years (2012–2022). Data accessed from National Library of Medicine (PubMed) on 9 March 2022.

**Figure 2 jfb-13-00081-f002:**
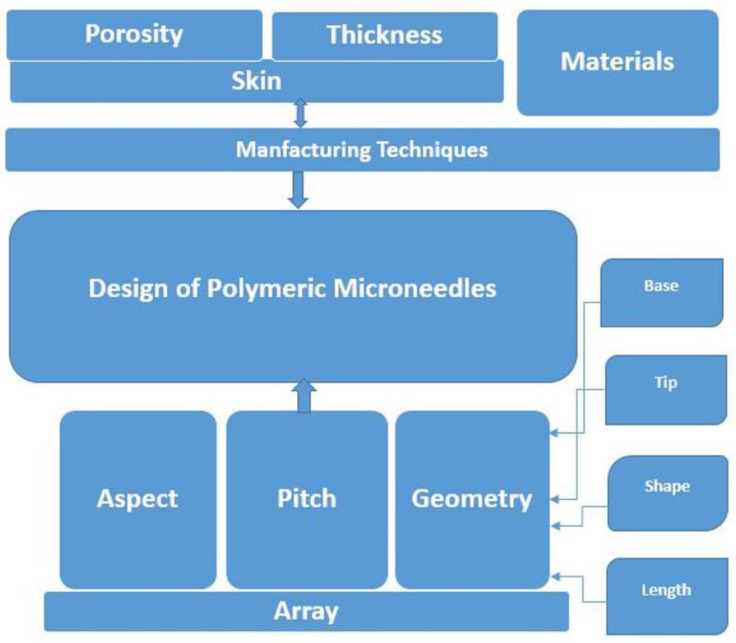
Basic parameters—design and performance of polymeric MNs.

**Figure 3 jfb-13-00081-f003:**
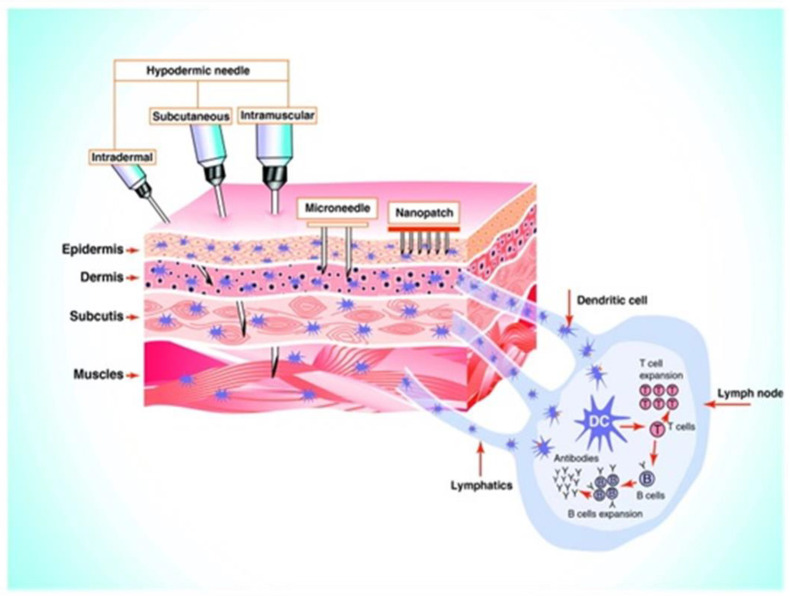
Graphical illustration of permeation of conventional hypodermic needle vs. MNs when inserted into the skin layers. The image is reproduced with permission from [60], Elsevier, 2011.

**Figure 4 jfb-13-00081-f004:**
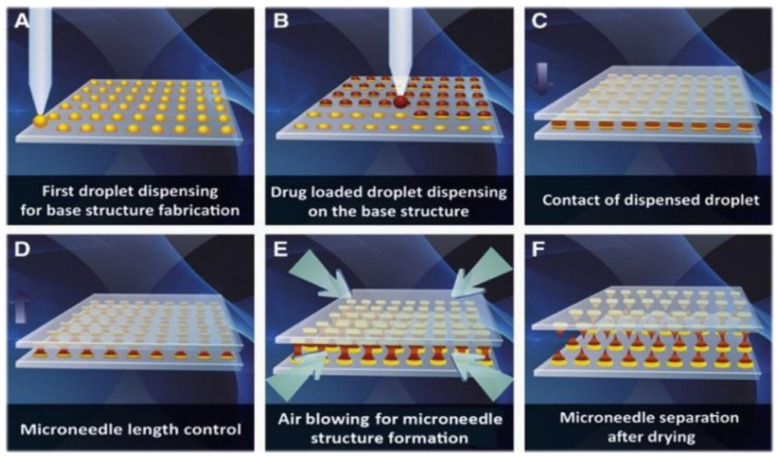
A graphical representation of the DMN manufacturing process, which employs the droplet air-born method. (**A**) Dispensing of biopolymers on a flat surface for the fabrication of a base structure. (**B**) Drug-containing droplets are dispersed across the base structure. The contact of dispensed particle droplets is caused by the downward movement of the upper plate. (**C**) The point of contact of the dispensed droplet. (**D**) Regulating microneedle length. (**E**) Droplet solidification mediated by air blowing to shape microneedle structure. (**F**) Plate separation dissolves microneedle arrays on the upper and lower plates. The image is adapted from [86], Elsevier, 2013. Droplet-born air blowing (DAB) technology for creating polymeric MNs.

**Figure 5 jfb-13-00081-f005:**
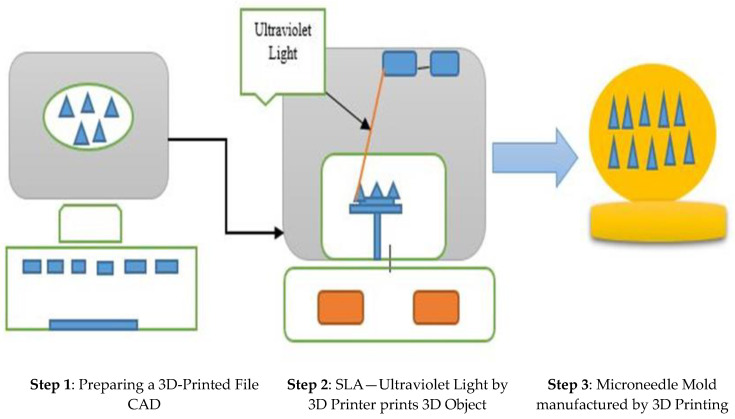
Graphical representation of 3D printing process of polymeric MNs.

**Figure 6 jfb-13-00081-f006:**
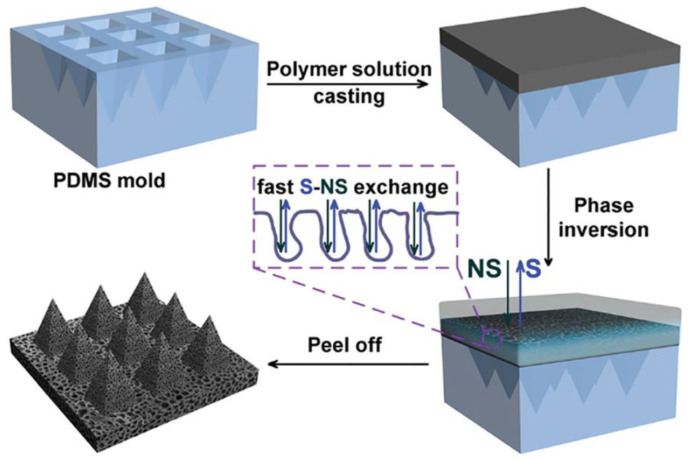
Schematic inversion fabrication technique for porous polymeric micro needles. Reproduced with permission from [45].

**Figure 7 jfb-13-00081-f007:**
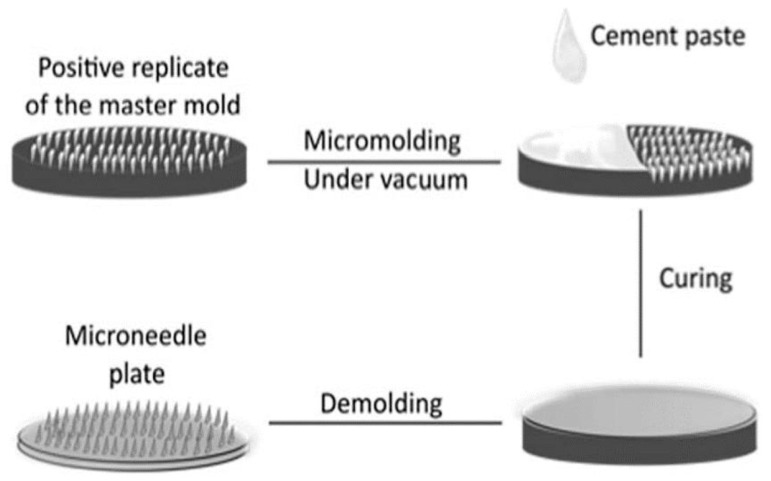
Schematic diagram of fabrication process for porous-based polymeric microneedles. Reproduced with permission from [46].

**Figure 8 jfb-13-00081-f008:**
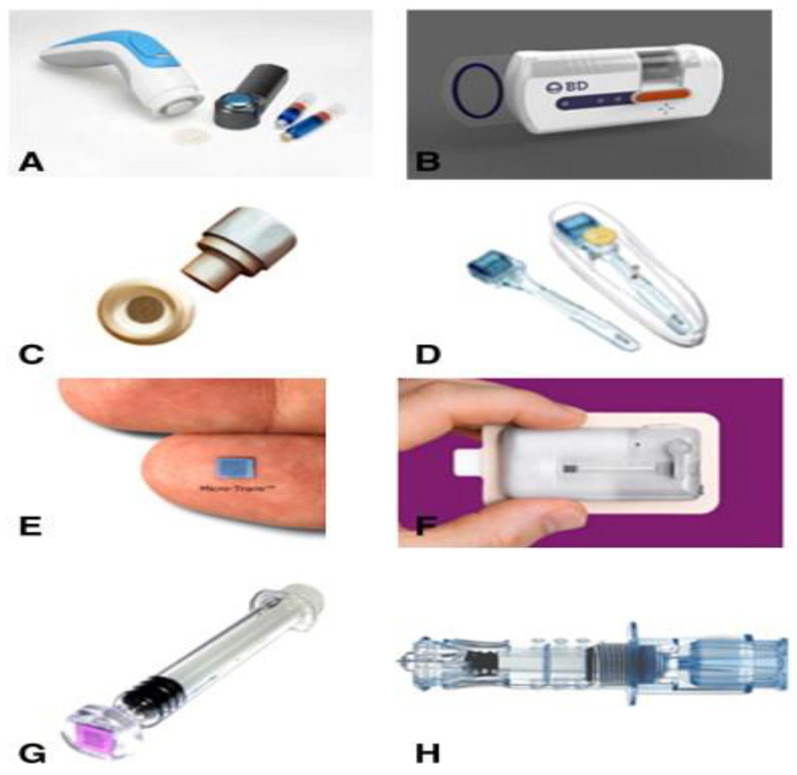
Current microneedle devices. (**A**) Microstructured transdermal system (MTS). (**B**) Microinfusor. (**C**) Macroflux^®^. (**D**) Microneedle Therapy System (MTS Roller™). (**E**) Micro-trans™. (**F**) h-patch™. (**G**) Micron Jet. (**H**) Intanza^®^. Reproduced with permission from [43].

**Figure 9 jfb-13-00081-f009:**
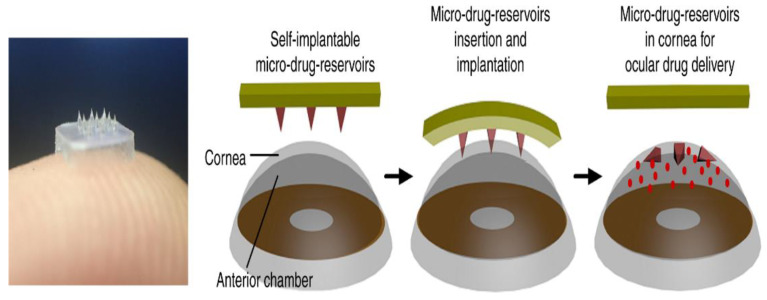
Eye contact path for ocular drug delivery. Reproduced with permission from [44] Nature Publishing Group.

**Table 1 jfb-13-00081-t001:** Typical polymers—general properties and fabrication of polymeric MNs.

Material	Advantages	Limitations	Fabrication Techniques	References
1. PVA	Low material costs Good plasticity Dissolvability and nontoxicity	Greater rate of absorbency	Molding fused deposition method (FDM)	[23,24]
2. PLGA	Preparation of dissolving microneedles (MNs)	Material costs are high	Molding, hot embossing	[25,26]
3. HA	Faster rate of dissolving	Chances of skin irritability	Micro-molding	[27]
4. PCL	Good thermal stability High rate of permeability	Process of slow degradation	3D printing, micro-molding	[28,29]
5. PEGDA	Can penetrate easily into molecular spaces	High material cost	Photolithography	[30]
6. PGA	Faster rate of degradation Exceptional mechanical strength	High material cost	Injection molding technique lithography	[31,32]
7. PLA	Higher rates of tensile strength Excellent physical and mechanical rigidity	Costly material Slower rate of degradation	Molding	[33,34,35]
8. PVP	Good plasticity and dissolvability	Difficulty in fabrication	Molding and photo polymerization	[36]
9. PDMS	Good biocompatibility and flexibility	Less penetrability	Micro-molding, curing	[37]
10. Poly (Ethylene Glycol-co-methacrylic-acid)	Good biocompatibility	Good drug transport mechanism	Bulk polymerization	[37]
11. Cellulose acetate	Good base material	Bio fluid extractions/insulin delivery	Mold casting method	[37]
12. PGMA	Good penetration efficiency	Drug delivery/ISF sampling	photopolymerization	[37]

Abbreviations: PVA: polyvinyl alcohol, PLGA: poly lactic-co-glycolic acid, HA: hyaluronic acid, PEGDA: poly (ethylene glycol) diacrylate, PGA: polyglycolide; PLA: poly (lactic acid), PVP: polyvinylpyrrolidone, PDMS, Poly (Ethylene glycol-co-methacrylic-acid), PGMA: Poly Glycidyl methacrylate.

## Data Availability

Data sharing is not applicable to this article as no new data were created or analyzed in this study.

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
