# Peer review of "The Rise of Polymeric Microneedles: Recent Developments, Advances, Challenges, and Applications with Regard to Transdermal Drug Delivery"

_jfb, 2022, doi:10.3390/jfb13020081_

Round 1

Reviewer 1 Report

In this review, Gera et al. introduced the polymeric microneedles in the terms of fabrication methods, challenges and applications. The manuscript can be accepted before addressing following issues:

  1. The authors should give a content of this review to make the review much clearer.
  2. The authors should indicate how Figure 1 is made (e.g., which keywords are used in exploring the publications) This information should also be included in the figure caption.
  3. The quality of the figures needs to be improved.
  4. For section 3.1, the enzyme responsive microneedle should be added and discussed.
  5. In the applications part, Figures should be added and discussed when give examples.
  6. More discussion and comparison should be added especially in introducing the fabrication of the microneedle and their application.

Author Response

Dear Reviewer, thank you for the suggestions and changes needed in the manuscript. The manuscript is modified accordingly, thank you. Please see the attachment

Reviewer 2 Report

  • I felt this paper is too generic and there is not much information about techniques or different polymers. Everything is very brief. Authors included mathematical modelling, but they didn't discuss it well. 
  • Line 155 Fig 1 was referred instead of fig 2. Please correct it to fig 2.
  • Table 1 the materials are very few and more common. Please try to include more.
  • Line 195 It should be epidermal thickness not “long”
  • Line 200 Fig 1 was referred again instead of fig 3
  • Figures 4 and 5 are not referred in the paper.
  • No discussions on different equipment & focused only on droplet- born air blowing.
  • Authors also should look for more recent papers and get more relevant information
  • Pharmaceuticals 202215(2), 190; https://doi.org/10.3390/ph15020190
  • https://doi.org/10.1002/adfm.202270059
  • https://doi.org/10.1016/j.ijpharm.2021.121295
  • https://doi.org/10.1016/j.mtbio.2022.100217

Author Response

Dear Reviewer, thank you for the suggestions and modifications needed in the manuscript, it was modified accordingly and sent to the Assistant Editor.

Round 2

Reviewer 1 Report

The work can be published in this journal in current form. 

Author Response

We want to thank the reviewer for your thoughtful comments and for helping us with the improvisation of the manuscript. 

The English language corrections are addressed and spell checks are checked accordingly. 

We are looking forward to your good selves in signing the review report.

Reviewer 2 Report

I appreciate the authors for revision version of manuscript.

I would like to suggest the authors to refer to recently published articles on microneedles and update the relevant matter with citations.

Microneedle-Based Natural Polysaccharide for Drug Delivery Systems (DDS): Progress and Challenges: https://doi.org/10.3390/ph15020190

The figure 2. represented can be in better form, please change figure 2

Authors should check the English formatting throughout the manuscript and chemical symbols represented: O2, H2O2

Author Response

We want to thank the reviewer for your thoughtful comments and for helping us with the improvisation of the manuscript.

  1. The English language corrections are addressed and spell checks are checked accordingly.
  2. Fig 1 - The title for the figure is addressed
  3. Fig 2 is changed totally with a new graphics
  4. Chemical symbols are addressed wherever necessary.

We are looking forward to your good selves in signing the review report.
